# Spatio-Temporal Assessment of Heterogeneity by Logging Intensity in a Federal Concession Area in the Brazilian Amazon

Afonso Henrique Moraes Oliveira [1,*], Lucas José Mazzei de Freitas [1,2], Mauro Mendonça Magliano [3], José Humberto Chaves [4], Carlos Tadeu dos Santos Dias [5] and Lucieta Guerreiro Martorano [1,2]

[1] Programa de Pós-Graduação em Sociedade, Natureza e Desenvolvimento, Universidade Federal do Oeste do Pará, Rua Vera Paz—Salé, Santarém 68000-000, PA, Brazil; lucas.mazzei@embrapa.br (L.J.M.d.F.); lucieta.martorano@embrapa.br (L.G.M.)

[2] Embrapa Amazônia Oriental, Belterra 68143-000, PA, Brazil

[3] Instituto Nacional de Criminalística, Polícia Federal, Brasília 70610-200, DF, Brazil; mauromagliano@gmail.com

[4] Serviço Florestal Brasileiro—SFB, Brasília 70818-900, DF, Brazil; jose.chaves@florestal.gov.br

[5] Departamento de Ciências Exatas, Escola Superior de Agricultura Luiz de Queiros, Avenida Pádua Dias, 235, Piracicaba 13400-000, SP, Brazil

[*] Correspondence: afonso.oliveira@discente.ufopa.edu.br; Tel.: +55-11-933-932-057

**Abstract:** The logging intensity often does not take into account the spatial heterogeneity of the forest volume of commercial native species in the Brazilian Amazon. This study aims to evaluate the spatio-temporal heterogeneity distribution by assessing logging intensity and its effects on the volumetric stock and abundance of commercial species, with a focus on sustainable management practices. This study was conducted in the Saracá-Taquera National Forest in the Brazilian Amazon. Forest inventory data, elevation, and PlanetScope satellite images were integrated into a geographic information system. The information was aggregated into regular 1-hectare cells for the times before, during, and after logging (t0, t1, and t2). The unsupervised classification algorithm k-means with four clusters was used to analyze heterogeneity. Before logging, areas with higher commercial volumes were distant from water bodies, while areas with lower elevation had lower wood stocks. Logging intensity was generally low, concentrating on a few trees per hectare. Logging in the study area revealed a heterogeneous spatial distribution by intensifying in areas with the highest wood stocks. These results suggest that, in addition to the recommended logging intensity according to legislation, forest heterogeneity should be considered by the manager, promoting adaptive strategies to ensure the conservation of forest resources.

**Keywords:** sustainable forest management; logging intensity; tropical forest; reduced impact logging; geoprocessing; volumetric heterogeneity

## 1. Introduction

Primary forests represent some of the most vital ecosystems on Earth [1]. Approximately 1.6 billion people rely directly on forests, with the forestry industry contributing approximately US $661 billion to the global GDP, and forests absorb an estimated 7.6 billion tons of $CO_2$ annually [2–5]. The Brazilian Amazon alone encompasses one third of the world's tropical forests, boasting commercial roundwood reserves totaling around 60 billion cubic meters, thereby establishing itself as the largest repository of tropical timber worldwide [6–9].

In the Amazon, owing to its high species diversity, there remain few studies on the volumetric distribution of the forest [10]. Despite extensive discussion among researchers on the topic of volume, there are still gaps in the findings concerning the spatial distribution of volume among native Amazonian species [10–16].

When observed at a synoptic scale, the terra firme Amazon Forest seems like a plain with homogeneous vegetation cover. However, local analysis unveils a diversity of environments influenced by factors such as topoclimatic and pedological conditions, as well as water availability and physiographic characteristics [17–20]. This typological heterogeneity significantly impacts the composition and volumetric distribution of vegetation in terra firme forests in Amazonia [21–23].

The joint use of remote sensing products and forest inventory data applied to forest management has been gaining ground [24–26]. As the topography of the site has a direct influence on the occurrence of species and most of them are distributed non-randomly, information from remote sensing, such as elevation, plays a crucial role in mapping and monitoring species [27–31]. Wolf [26] assessed the spatial distribution of species richness in tropical forests using lidar data and found significant differences in richness associated with topographic variation. Dong [32] used optical images from the Landsat series together with topographic correction models and forest inventory data to classify groups of forest species according to topography and obtained satisfactory results.

Remote sensing can also play an important role in monitoring and quantifying canopy disturbance caused by selective logging [4,16,24–26]. Fortunately, damage to the canopy is highly correlated with the volume of timber removed from the forest [24,27]. The disturbances caused by logging also vary depending on the logging practice used; the recommended logging practice in the Amazon is known as reduced impact logging (RIL), and should be incorporated into management plans to minimize impacts [27].

Previous studies have shown the need for high spatial and temporal resolution images for monitoring selective logging in the Amazon [26,32,33]. In addition, most of the remote sensing techniques and products used for mapping and monitoring studies of selective logging have been insufficient for large-scale assessments [24]. A comparison using field data of crown disturbance with satellite imagery from the Landsat series after logging proved that traditional analytical methods and medium spatial resolution imagery fail to detect around 50% of the crown damage caused by forest harvesting operations [27]. Abdollahnejad [24] proposed an advanced approach integrating geographic information systems (GISs) and remote sensing using very high spatial resolution images to monitor logging areas and pointed to an increase in the accuracy of volume estimates as a function of the spectral and spatial resolution of the images. Petri [34] tested the use of images from the PlanetScope nanosatellite constellation for vegetation studies in the Amazon and concluded that high spatial and temporal resolution images are fundamental for understanding forest dynamics in the Amazon. Sustainable forest management (SFM) is recognized as a strategy and indicator of forest conservation, but there are gaps in the appropriate conditions for exploiting forest resources [35,36]. Brazilian standards establish specific values for harvesting intensity per hectare. On the other hand, when assessing sustainability indicators, factors such as the distribution of the diameter class structure and the availability of species to constitute the cutting rate in the forest management process must be considered [37].

The authorized intensity of logging in tropical forest management is not associated with data on the heterogeneity of the original forest structure; that is, the volumes determined for extraction are fixed and standardized [38]. Carvalho considered the difficulty of tropical forest management due to the complexity and heterogeneity of its ecosystems [39]. According to Chazdon, the inappropriate use of natural forests can disrupt extraction cycles and degrade ecosystems [40].

Cutting intensity is one of the most important aspects of forestry and refers to the commercial volume of the trees to be harvested, estimated using volumetric equations provided in the sustainable forest management plans (SFMPs) and based on data from the pre-harvest forest inventory, expressed in cubic meters per unit area (m$^3$ ha$^{-1}$). Currently, Brazilian legislation authorizes a maximum cutting intensity of 30 m$^3$ ha$^{-1}$ for 35-year cutting cycles [41,42].

Although Brazilian legislation establishes a maximum limit for logging intensity, this fixed limit is indiscriminately applied to the entire Annual Production Unit (APU) area, disregarding possible forest heterogeneity. This becomes a sensitive issue as managers can exploit the forest without respecting its original structure and spatial distribution [43,44]. Therefore, the objective of this study is to evaluate the spatio-temporal heterogeneity by logging intensity in a federal concession area in the Brazilian Amazon within an SFM area following logging activities subjected to the reduced impact logging (RIL) technique.

## 2. Materials and Methods

### 2.1. Study Area

The study was carried out on a federal forest concession area in the Brazilian Amazon, situated within the Saracá-Taquera National Forest in the western part of the state of Pará, covering an area of 441,152 hectares (Figure 1). This sustainable Use Conservation Unit (UCS) was established by Decree No. 98.704 on 27 December 1989, with the objective of conducting research projects and initiatives aimed at the sustainable utilization of forest resources and the wellbeing of the populations residing within the UCS. These conservation units (UCs) were established under the National System of Nature Conservation Units and managed by the Chico Mendes Institute for Biodiversity Conservation (ICMBio) [44].

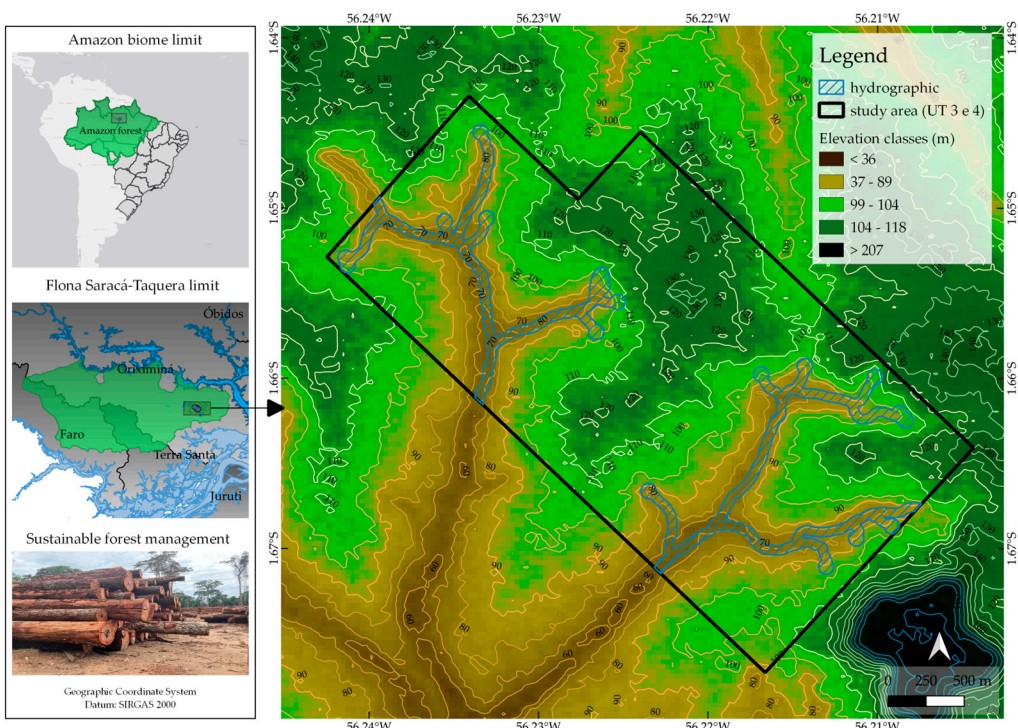

**Figure 1.** Location of the study area.

The analyses were carried out in the eastern section of the national forest, specifically within Forest Management Unit II (FMU II). In SFM, the FMU represents the designated portion of the property allocated for forest management. The specific area designated for logging activities is referred to as a timber unit (TU). The portion of forest slated for management each year is termed an annual production unit (APU), and an APU may encompass one or more TUs.

The APU under evaluation was harvested between May 2022 and May 2023, with an initial authorization to exploit 1629 hectares and a total roundwood volume of 39,031 m$^3$, comprising 30 commercial species selected by the concessionaire. TUs 3 and 4 of APU 11 were chosen to test the hypothesis proposed in this study. TU 3 covers an area of 339 hectares, while TU 4 covers 369 hectares, resulting in a total effective study area of

735 hectares, which also includes areas forming part of the hydrographic network in the analyzed area (Figure 1).

## 2.2. Vegetation

The national forest is covered by tropical rainforest, with variations typically linked to geomorphological features. The regional vegetation can be categorized as submontane and lowland ombrophilous dense forest, distinguished by two distinct strata; one emergent, featuring *Dinizzia excelsa*, *Bertholletia excelsa*, and *Cedrelinga catanaeformis* as primary species, and the other uniform, marked by the presence of *Manilkara* spp., *Protium* spp., and *Pouteria* spp. [45].

The two primary facies, submontane and lowlands, comprise 94.1% of the Flona's area, while pioneer formations influenced by rivers account for 2.7%, and campinarana for 0.2%. Primary natural vegetation formations constitute 97% of the Flona's vegetation cover, whereas areas affected by anthropogenic activities represent 2% of the national forest's total area [46].

## 2.3. Forest Inventory Data Extraction

The data from the pre-harvest inventory of commercial tree species and the inventory of harvested trees were obtained from the Brazilian Forestry Service (BFS) in the form of an electronic spreadsheet in .xlsx format. The BFS oversees concessions for sustainable forest management in public forests in the Amazon. Additionally, the BFS provided the post-harvest forestry report for APU 11, which included geospatial data on all inventoried commercial trees, along with an extra column containing the date of tree felling. Spatio-temporal analyses were conducted based on the 'cutting date' information to compare the original forest structure, the logged structure, and the structure remaining after logging.

Under the New Forest Code (Federal Law 12.651/12), harvesting trees within permanent preservation areas (PPAs), such as those near watercourses, is prohibited. Consequently, trees located within PPAs were excluded from the analysis to focus solely on potential trees for harvesting (Figure 2). The legislation also mandates a minimum cutting diameter (MCD) of 50 cm, meaning trees smaller than this diameter would not be felled and were thus excluded from the analysis as well. Pre-harvest inventory requirements stipulate that trees with a minimum diameter at breast height (DBH) of 10 cm below the MCD should be included [44]. Thus, the study focused on evaluating only commercial trees available for cutting outside the limits of PPAs and with a DBH greater than 50 cm.

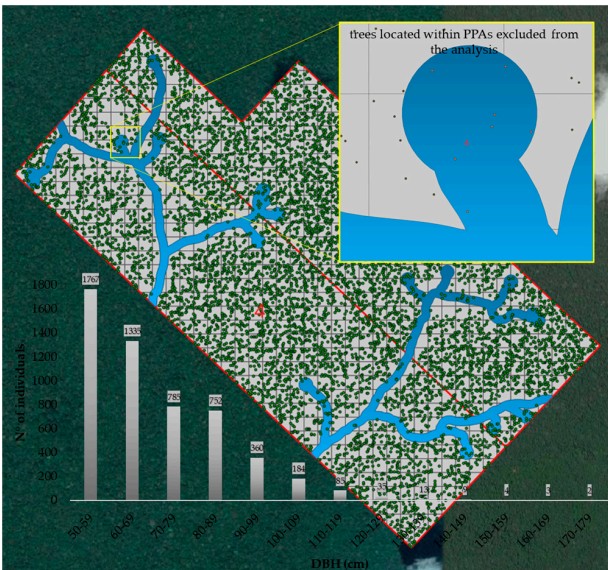

**Figure 2.** Spatialization of the inventory trees (points in green) with a focus on the exclusion of areas within PPAs (in blue) and the diametric distribution graph.

### 2.4. Measuring Topographical Variables

The topographic variable of elevation (m) was utilized, based on the hypothesis that this variable can influence the spatial distribution of commercial species volume [23–26]. Elevation was calculated using the average altitude for each 1-hectare cell. The Copernicus Global DSM Digital Elevation Model (DEM) at 30 m resolution was employed. The data have an absolute vertical accuracy >4 m (90% linear error) and absolute horizontal accuracy >6 m (90% linear error). These data originated from the TanDEM-X mission between 2011 and 2015 and were made available for free use in 2019. They are widely used in research employing the approach utilized in this work [47,48].

### 2.5. Assessment of Forest Canopy Openings Caused by Selective Logging

To evaluate the gradient of forest canopy openings resulting from tree felling in relation to logging intensity, satellite images from the PlanetScope constellation were employed. A pair of images before and after the exploitation were used for change detection analysis. This was based on the premise that higher logging intensities lead to larger openings in the forest canopy, thereby resulting in variations in the gradient of forest cover changes [24,27,49].

The PlanetScope constellation comprises multiple launches of individual satellite groups (DOVEs), each consisting of a constellation of 3U CubeSats (i.e., $10 \times 10 \times 30$ cm) with over 120 active DOVEs. These sensors operate in at least four spectral bands; blue (455–515 nm), green (500–590 nm), red (590–670 nm), and near-infrared (780–860 nm), offering 3 m of spatial resolution and 12 bits of radiometric resolution [34]. Band 3 was utilized individually due to its heightened spectral response to exposed soil and dry vegetation targets, making it highly recommended for vegetation studies in the Amazon [50–52] (Figure 3).

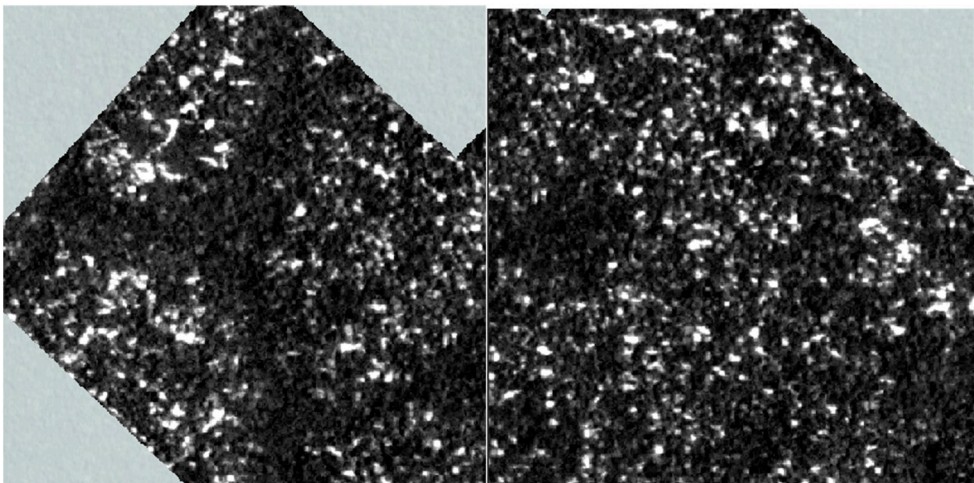

**Figure 3.** In the lighter tones of the band 3 images, there are higher spectral response values for areas with exposed soil and dry vegetation (clearings).

Figures 4 and 5 illustrate, didactically, the process of harvesting a tree and the impact its toppling can have on the forest canopy, resulting in the opening of a clearing. Thus, based on the variations in signal intensity captured by the sensor and the frequency histogram of the image using band 3, a threshold was empirically defined based on the RGB (3,2,1) composition to classify areas with and without change. Digital number (DN) values greater than 2550 were considered canopy, and DN values less than 2550 were considered gaps.

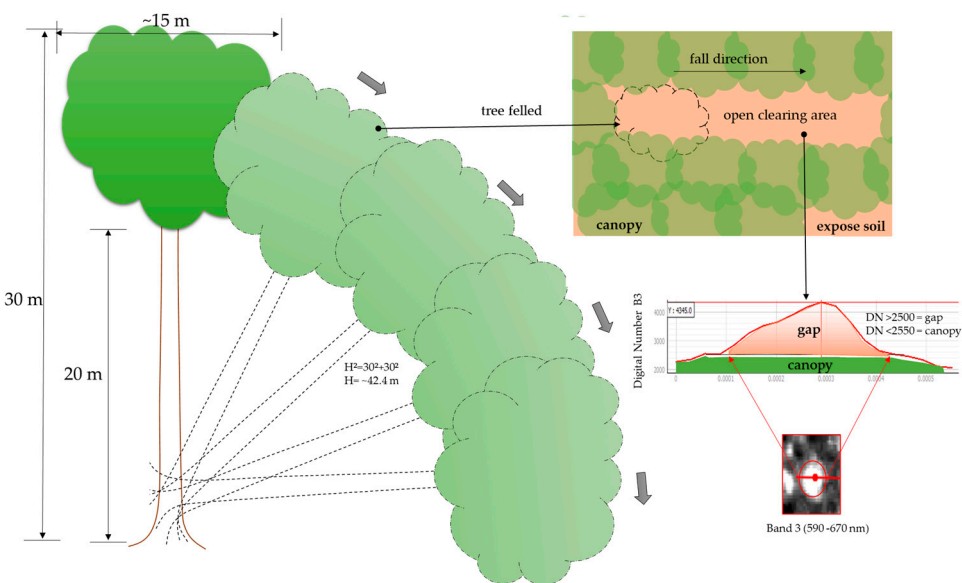

**Figure 4.** Illustration of the felling of a tree and its impact on the forest cover.

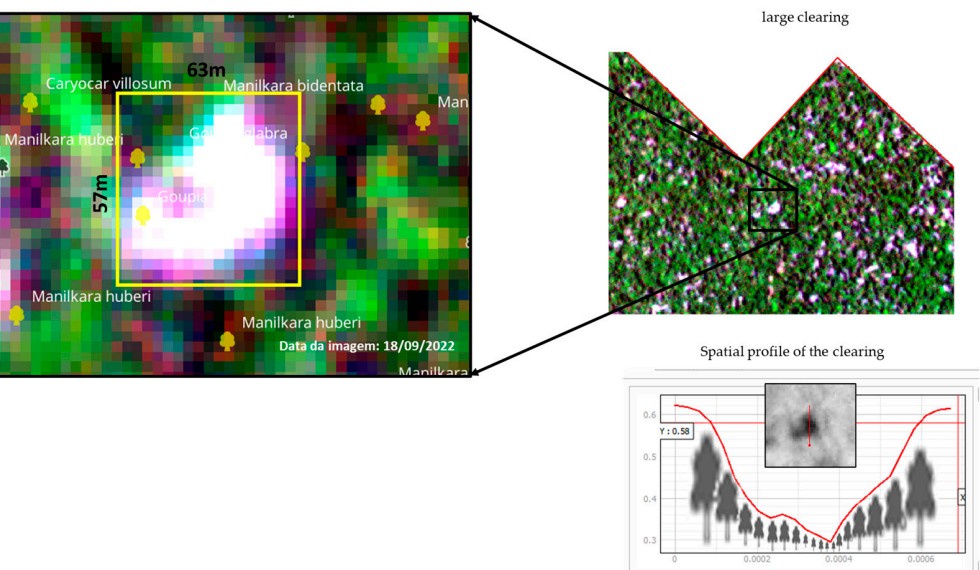

**Figure 5.** Empirical analysis of the size of the glades as a function of the areas extracted.

After classification, binary thresholding or binarization of the image in raster format was performed to separate areas of dense forest from other areas (such as exposed soil and dry vegetation, among others). Following binarization, the forest and clearing classes were analyzed by converting the file to vector format. Only the clearings were then extracted to calculate the area of each class, excluding individual 3 m × 3 m (9 m$^2$) pixels. The data were subsequently verified through visual interpretation. For visual analysis, we used the coordinates of all the trees that had been logged and the result of the classification of the clearings, so that for each tree or group of trees logged, a polygon was assigned and classified as a clearing.

Thus, except for the individual pixels, all other data points were validated and aggregated into 1-hectare cells. The metrics employed to aggregate the clearing vectors into 1-hectare cells included the percentage of the area within the cell and the total clearing area. Linear correlation analysis was conducted to assess the correlation between the area of clearings and logging intensities.

### 2.6. Analysis of Changes in the Spatial Distribution of Tree Volume and Abundance

The inventory information was spatialized and aggregated into 1-hectare cells. This decision was made for two main reason; firstly, to enable the application of zonal statistics, and secondly, to facilitate understanding and discussion of the information. Since 1 hectare is a widely used unit of area in discussions of this nature, it allows for easier understanding, interpretation, and comparison between different studies and regions [33,53–55].

The data were spatialized considering three time classes. Class "t0" refers to all the individuals inventoried before logging, class "t1" refers to the trees that were actually extracted from the forest, and class "t2" refers to the trees remaining in the area after logging. The diagram shown in Figure 6 illustrates the aggregation of individuals in the 1-hectare cells. For each cell, in each time class, the values contained in the inventory were assigned a tree abundance (n° ind. ha$^{-1}$) and volume (m$^3$ ha$^{-1}$) (Figure 6).

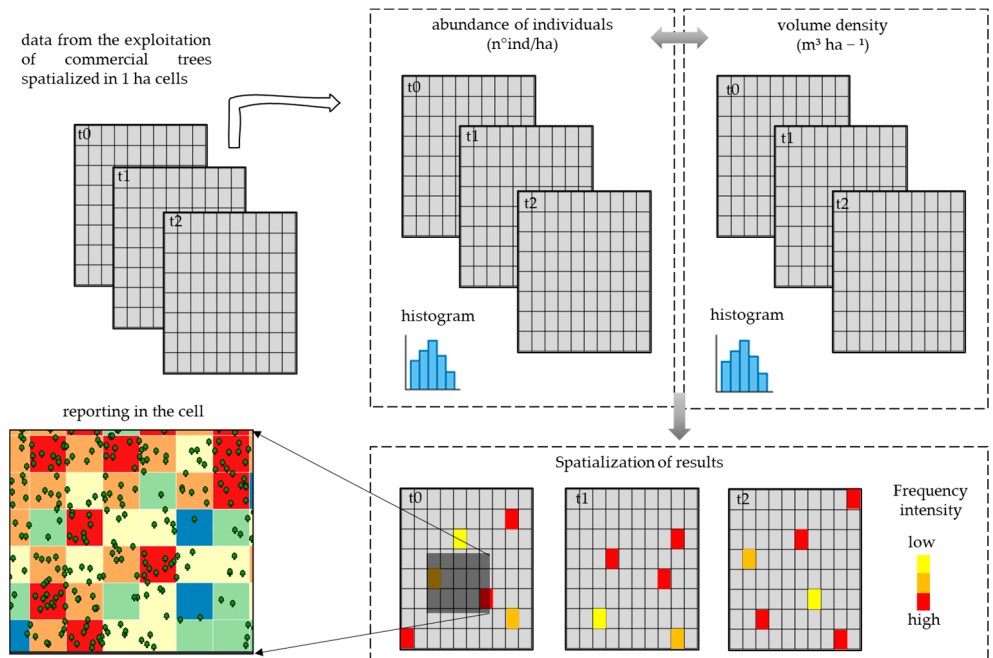

**Figure 6.** Schematic drawing of the aggregation of trees into 1 ha cells.

### 2.7. Determining the Number of Clusters and Analyzing the Spatial Distribution of Logging

Cluster analysis was conducted using the k-means algorithm to classify the area into groups based on the topographic variable and the volumes of timber extracted (t2). The algorithm identifies objects or entities with similar characteristics, creating groups or classes with high internal homogeneity (within clusters) and high external heterogeneity (between clusters) [56–59].

To determine the optimal number of clusters (k), the elbow method was employed through visual analysis of the graph. Utilizing the elbow method with k-means clustering in the context of forest analysis is a conventional approach in data science and ecology [60–64]. An evaluation of the dissimilarity of the variables (SSEs) among each cluster was conducted to identify potential heterogeneities in forest exploitation. At the conclusion of the process, 746 1-hectare cells (100 m × 100 m) were considered.

### 3. Results

#### 3.1. Analysis of the Change in the Spatial Distribution of Tree Volume and Abundance Due to Logging Activity

The spatial distribution of commercial volume before logging indicates that the interval between 45 and 60 m$^3$ ha$^{-1}$ had the highest number of cells with significant volume (199). For the higher volume ranges, above 120 m$^3$ ha$^{-1}$, only five cells were observed. It should

also be noted that the lowest commercial volume ranges, from 0 to 15 $m^3$ $ha^{-1}$, are mostly located close to water bodies (Figure 7). In terms of tree abundance, the intermediate ranges of commercial tree abundance of 5–10, 10–15, and 15–20 individuals per hectare concentrate the largest number of cells, with the range of 10 to 15 individuals being the most representative, with 306 cells counted, and consequently, the range with the largest number of total trees.

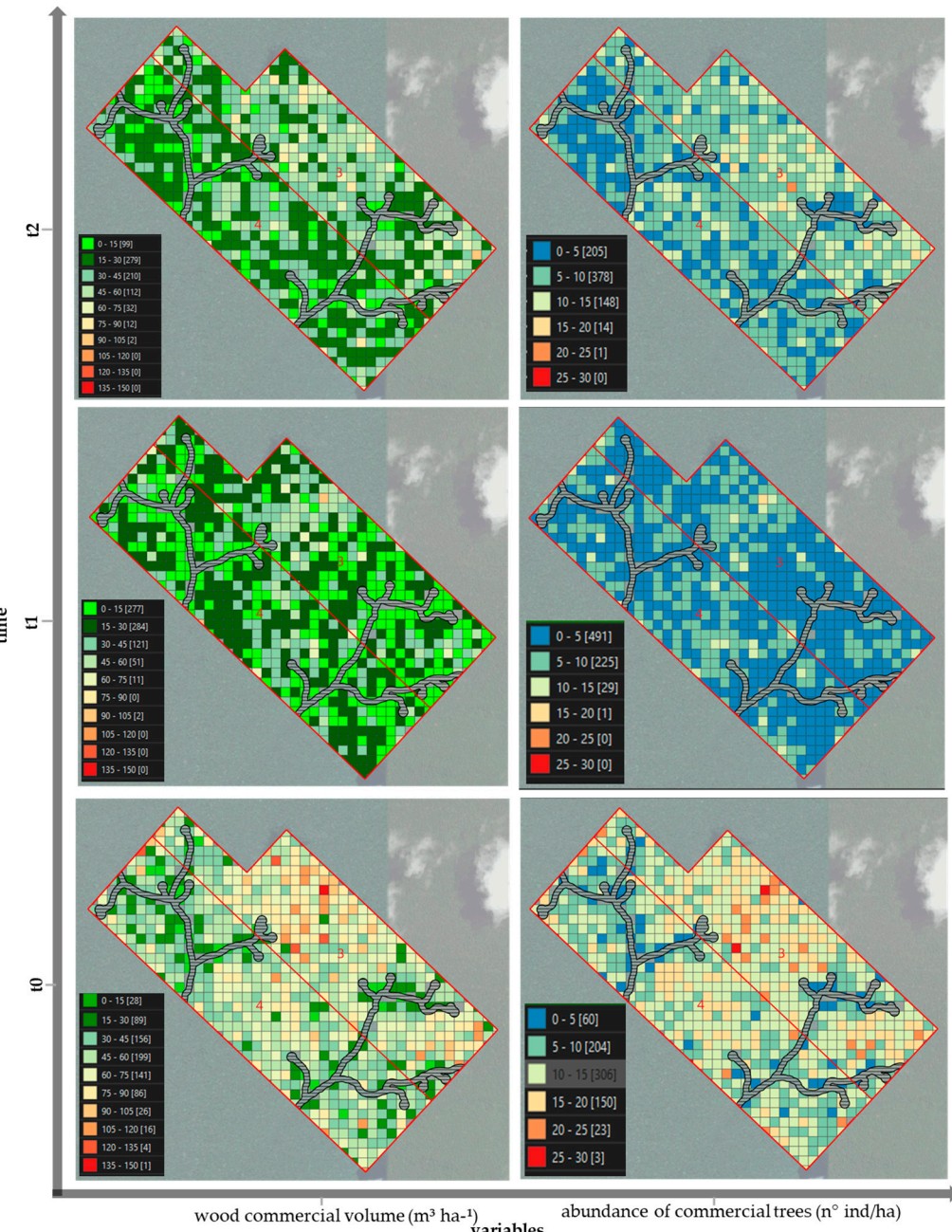

**Figure 7.** Spatial distribution of commercial volume and tree abundance before (t0), during (t1), and after logging (t2).

There was a gradient in the spatial distribution of both the commercial volume and the abundance of trees in the area before logging, with three cells having an abundance between 25 and 30 individuals per hectare and 60 cells with an abundance between 0 and 5 individuals per hectare. The range of variation in the stock of commercial volume

before logging is also extremely high, with 117 cells below 30 m$^3$ ha$^{-1}$ and 21 cells between 105 and 150 m$^3$ ha$^{-1}$ (Figure 7).

Regarding the remaining volume in the area, for didactic and visual purposes, it was decided to use shades of green to represent the volume bands below 30 m$^3$ ha$^{-1}$, which is currently the maximum intensity of exploitation per hectare stipulated by the legislation. Thus, it can be observed that before logging, the forest exhibited a wide distribution of cells in bands greater than 30 m$^3$ ha$^{-1}$ (Figure 7). After logging, the range between 15 and 30 m$^3$ ha$^{-1}$ concentrates the largest number of cells (279), suggesting a systematic reduction across the area, but leaving areas with a considerable volume of stored wood, such as areas with a volume of 75 m$^3$ ha$^{-1}$ or more, where 14 active cells remained. The map depicted in Figure 7 illustrates that the spatial distribution patterns of abundance and volume are similar throughout the area, both before (t0) and after logging (t2) (Figure 7).

After logging (t2), there was no significant change in the normal distribution curve of individual abundance, suggesting that the area was logged following the natural spatial distribution of the forest. However, a shift in the number of cells towards lower abundance ranges can be observed, with the range between 5 and 10 being the most representative after logging (Figure 8).

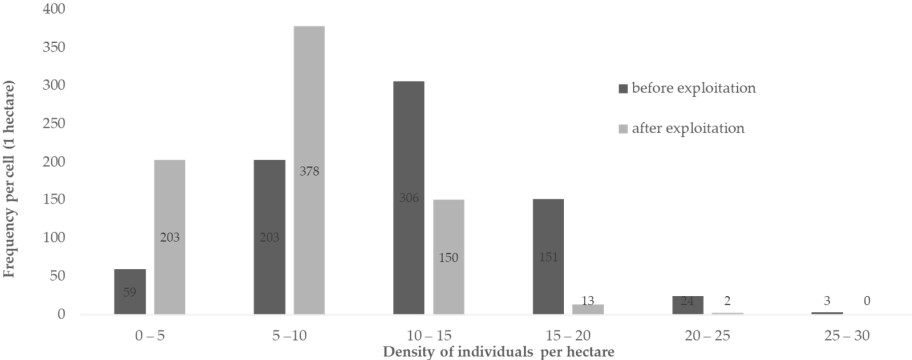

**Figure 8.** Frequency distribution histogram of the occurrence of 1-hectare cells in the abundance classes of commercial trees before and after logging.

While before logging, the largest number of cells was in the commercial volume range between 45 and 60 m$^3$ ha$^{-1}$, after logging, the class with the largest number of cells was concentrated in the range between 15 and 30 m$^3$ ha$^{-1}$, indicating the migration of cells to lower volume classes. However, it is still possible to identify areas with a high-volume stock, such as areas with a volume greater than 75 m$^3$ ha$^{-1}$, which represented 14 cells. The average volume per hectare decreased from 48 m$^3$ ha$^{-1}$ to 28 m$^3$ ha$^{-1}$, and the average number of individuals decreased from 12 to 7, considering the time before and after logging (Figure 9).

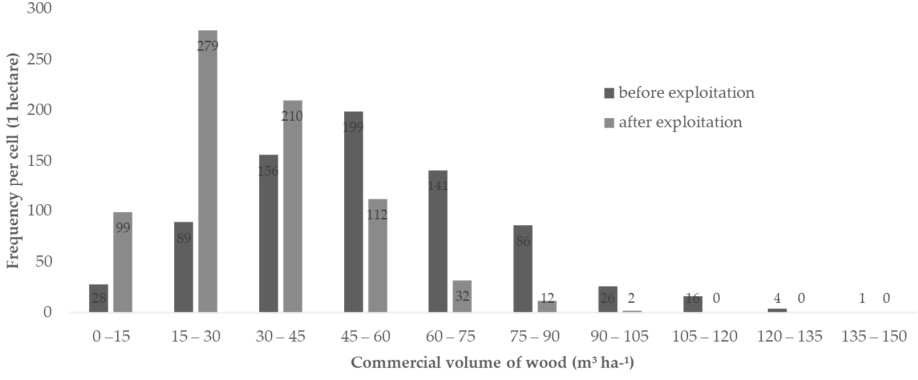

**Figure 9.** Frequency distribution histogram of the occurrence of 1-hectare cells in the commercial wood volume ranges before and after harvesting.

### 3.2. Analysis of the Harvested Trees

Assessing the results of the commercial trees actually harvested, it is evident that harvesting was concentrated on felling a few trees per hectare, with 65.5% of the cells showing the harvesting of between 0 and 5 individuals per hectare. However, 29 cells had a logging intensity between 10 and 15 individuals per hectare, and only one cell had a logging intensity above 20 individuals per hectare, as shown in Figure 10.

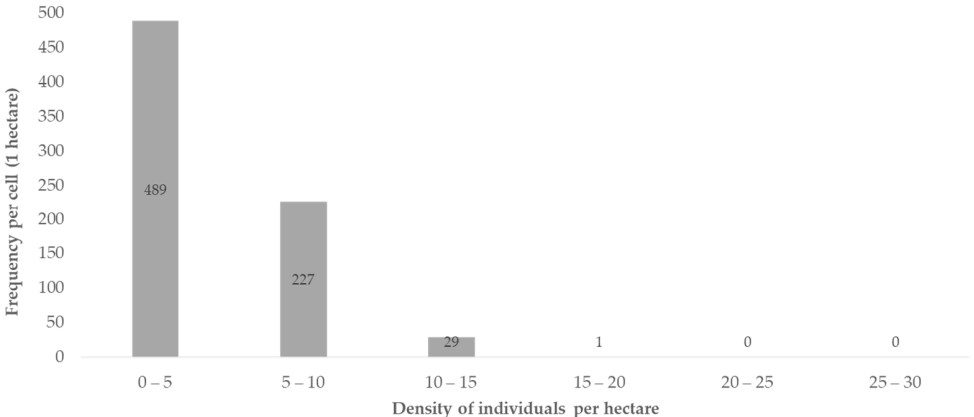

**Figure 10.** Class frequency distribution histogram of the number of trees logged per 1-hectare cell.

Examining the distribution map of the abundance of commercial trees logged in Figure 7 (t1), it is evident that most of the logging was concentrated on felling a few individuals per hectare, with more logging occurring in areas with a larger stock of trees, particularly in the plateau areas farthest from the water bodies.

In terms of the volume that was effectively logged, the highest number of sensitized cells was in the range between 15 and 30 $m^3$ $ha^{-1}$, comprising 284 cells (Figure 11). It was observed that approximately 75% of the cells were below 30 $m^3$ $ha^{-1}$, which is the maximum volume allowed by law. However, it should be noted that the remaining 25% were exploited, showing harvesting above 30 $m^3$ $ha^{-1}$, with two cells having an exploitation intensity between 90 and 105 $m^3$ $ha^{-1}$.

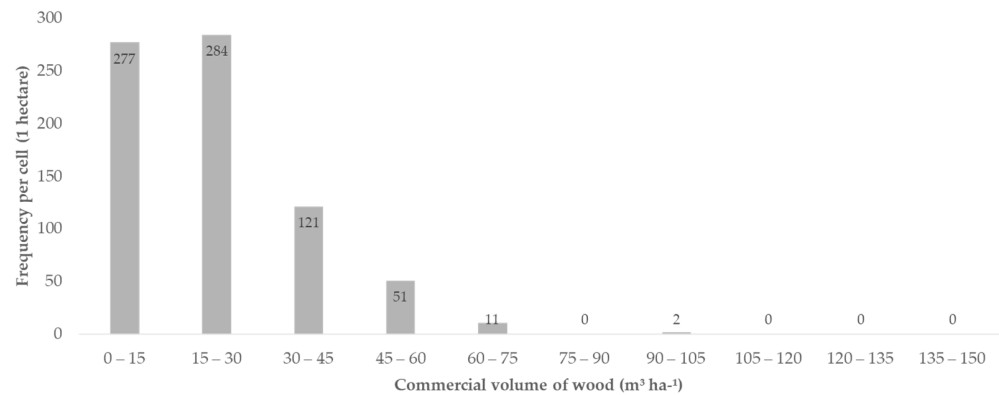

**Figure 11.** Frequency distribution histogram of logging intensity classes ($m^3$ $ha^{-1}$).

### 3.3. Digital Elevation Model and Its Relationship with the Distribution of Harvested Commercial Timber Volume

Elevation ranged from 50 to 200 m. The highest values for the volume of harvested commercial timber were found away from the water bodies and close to the plateau areas (48.6 to 98.4 $m^3$ $ha^{-1}$). The lowest volume values are mostly concentrated in the lower elevation regions, such as the water bodies present in the area (0 to 8.9 $m^3$ $ha^{-1}$) (Figure 12).

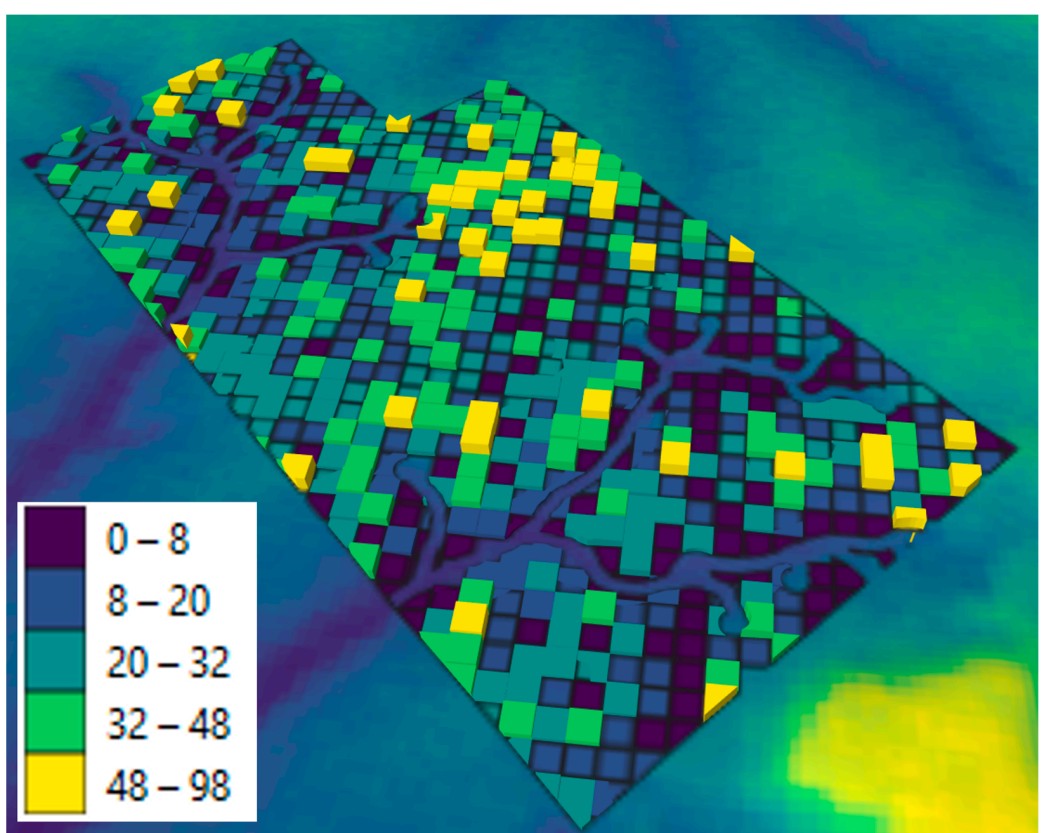

**Figure 12.** Relationship between elevation (m) and the volume of commercial timber actually harvested (m$^3$ ha$^{-1}$).

*3.4. Digital Elevation Model and the Relationship with the Distribution of the Volume of Commercial Timber Harvested*

Based on the slope of the distortion curve in relation to the number of clusters, four clusters (k) were defined for the data set (Figure 13).

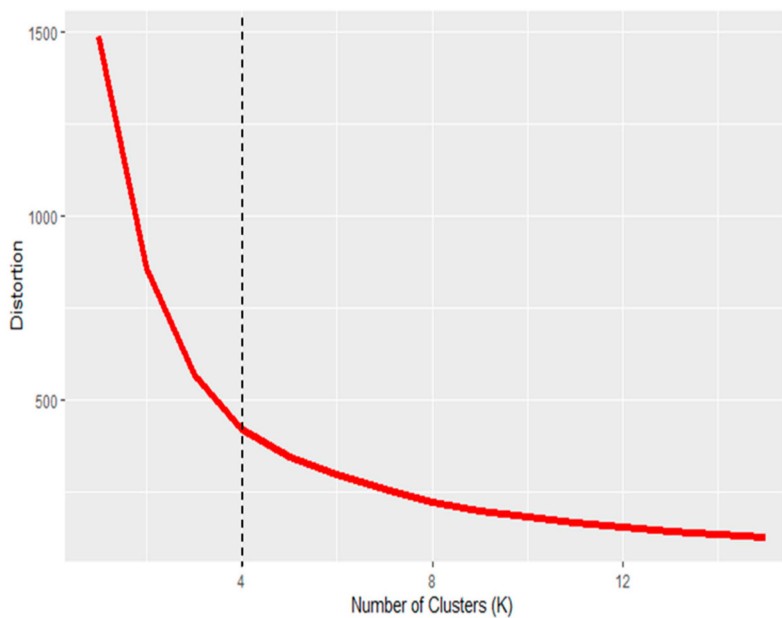

**Figure 13.** Definition of the best number of groups (k) using the elbow method.

Cluster 1 (C1), comprising 240 observations and representing 32% of the data, has average values of 14.7 m$^3$ ha$^{-1}$ and 113 m for the volume exploited and elevation, respectively, making it a transition group between low and high relief areas. Cluster 2 (C2) includes 220 observations (29%), with an average of 26.4 m$^3$ ha$^{-1}$ and 97.6 m for volume and elevation, respectively. Cluster 3 (C3), with 167 observations (22%), had volume and elevation values of 5.9 m$^3$ ha$^{-1}$ and 84.7 m, respectively, indicating a group with a spatial association with water bodies. Cluster 4 (C4) had 119 observations (16%) with volume and elevation values of 46.6 m$^3$ ha$^{-1}$ and 115.2 m, respectively, indicating a spatial association of high-volume extraction in a plateau region (Table 1).

**Table 1.** Results of the cluster analysis in relation to volume extracted and elevation.

| Cluster Centers | | | | |
|---|---|---|---|---|
| **Cluster** | **Proportion (%)** | **Volume Explored (m$^3$ ha$^{-1}$)** | **Elevation (m)** | **SSE** | **Cluster Name** |
| C1 | 32.17% | 14.70 | 113.30 | 132.30 | Low volume, high elevation |
| C2 | 29.49% | 26.42 | 97.64 | 105.60 | Medium volume, medium elevation |
| C3 | 22.39% | 5.90 | 84.79 | 77.58 | Low volume, low elevation |
| C4 | 15.95% | 46.66 | 115.29 | 103.78 | High volume, high elevation |

Regarding the dissimilarity of the groups, expressed through the sum of squares within the clusters (SSEs), the cluster with the least variance in the Euclidean distances is the "low volume low elevation" group, with the lowest SSE value of 77.5. This indicates that in these areas, close to the water bodies, logging occurred in a homogeneous manner and with few individuals per hectare (Figure 14). On the other hand, the most heterogeneous group was the "low volume high elevation" group, with an SSE value of 132. Since this group has the largest number of cells (240) distributed throughout the area, it naturally exhibits a high degree of heterogeneity among its internal observations. The "medium volume medium elevation" and "high volume high elevation" groups have intermediate SSE values, indicating a medium degree of heterogeneity.

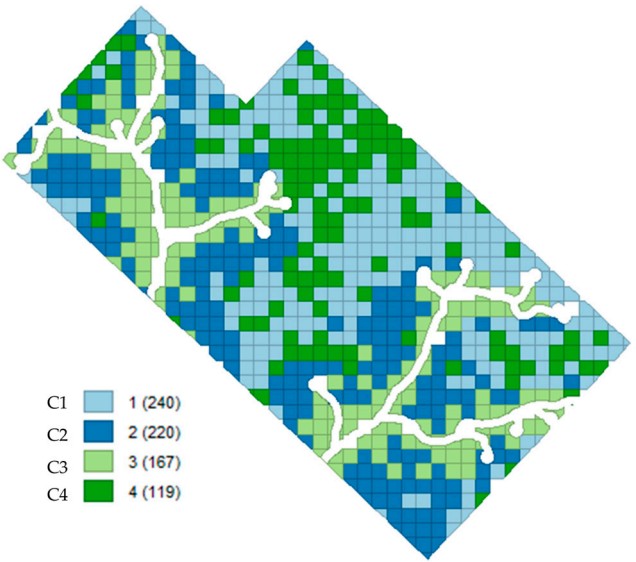

**Figure 14.** Spatial distribution of logging clusters using the k-means method.

*3.5. Evaluation of the Relationship between Canopy Openness and Selective Logging Intensity*

Figure 15 depicts the correlation graph between the volume of timber extracted (m$^3$ ha$^{-1}$) by intensity class and the area of the mapped clearing. A strong correlation

($r^2 = 0.93$) is observed, confirming the hypothesis of a positive relationship between the volume of wood extracted from the forest and the area of the clearing. In other words, this suggests that there is a correlation between the spatial and spectral information extracted by SR, representing the canopy opening resulting from logging activity.

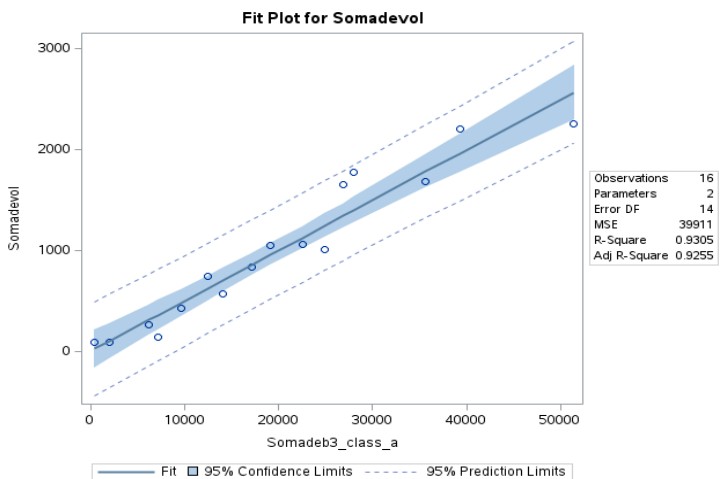

**Figure 15.** Positive correlation between the volume of timber extracted and the areas of clearing mapped.

## 4. Discussion

The main finding of this study demonstrates that the logging intensity in SFMPs using RIL techniques in the Amazon is not uniform and follows the natural heterogeneous distribution of the stand, with high logging intensity occurring in highly stocked areas. Despite the legislation setting a maximum cutting intensity of 30 $m^3$ $ha^{-1}$, there are areas within the work unit that have values above this limit, and areas that have intensity values lower than this limit. However, the study indicates that areas highly stocked with timber also exhibit higher logging intensity. One of the cells with high logging intensity (95 to 105 $m^3$ $ha^{-1}$) was also the cell with the highest available volume (120 to 135 $m^3$ $ha^{-1}$).

The results suggest the need to monitor forest recovery in areas with high logging intensity, to allow for the revision of forest management guidelines. This revision should consider the heterogeneous distribution of forest stands and adjust logging intensities according to the available wood stock in each management area [40,65]. The identification of non-uniform patterns in logging intensity indicates the necessity of developing adaptive management strategies that can be adjusted based on the specific characteristics of each management area [66,67]. Understanding the heterogeneous natural distribution of forest stands and their relationship with logging intensity can contribute to promoting the sustainability and conservation of forest resources by enabling more efficient and sustainable forest management [35,68,69].

These findings could inform new discussions regarding how legislation authorizes cutting intensity in management areas. By setting the intensity at 30 $m^3$ $ha^{-1}$, the legislation does not consider the spatial heterogeneity of the distribution of forest volume evidenced in this study [35,70,71]. The uniform application of this limit could lead managers to overexploit areas with low volume and underexploit areas with high stock. The decision to harvest is influenced by factors such as the species present, topography, or market demand for specific species or wood types. These factors can interact in complex and dynamic ways, influencing the logging strategies and intensities adopted by forest managers [72,73]. The results suggest that, in addition to the criteria adopted today, such as rarity and maximum volume per authorized area, among others, the inclusion of a criterion for maximum volume or maximum number of individuals exploited per hectare could be considered, taking into account spatialization on a more refined scale, rather than for the authorized area as a whole, as is currently conducted [44]. Currently, in Brazil, these criteria are defined by the Ministry of the Environment [41].

As demonstrated in this study, logging intensity is directly correlated with the available wood stock in specific areas, and is more intense where stock is higher and less intense where it is lower [24,38]. These findings are crucial for gaining a clear understanding of the spatial distribution of logging intensity in the Amazon and for guiding future enhancements in forestry legislation [74]. The results indicate that after logging, the spatial distribution of remaining volume and abundance per hectare remains proportionally similar to the original forest. It should be noted, however, that high logging intensities can leave the forest more vulnerable to collateral damage, such as forest fires [75].

The heterogeneity of logging was evidenced by several factors contributing to the spatial scale. Firstly, the topography of the study area varies between plateau areas with altitudes of 200 m and lowland areas with altitudes of 50 m. It is apparent that areas farther from water bodies had the highest concentration of logging, while in areas of lower altitude and those closer to water bodies, where it is likely more difficult for heavy machinery to operate, logging was less intense. Our classification was able to clearly define the relationship between average logging intensity (28 m$^3$ ha$^{-1}$) and elevation. The spatial distribution of commercial species and their preferences for different types of soil and terrain should be considered in further analyses.

The selection of commercial species for exploitation is determined by commercial interests rather than their spatial distribution characteristics, highlighting the importance of further research in this direction. This approach can lead to unsustainable logging practices and negatively impact the genetic diversity of species populations if these factors are not considered when deciding which trees to harvest. The polycyclic forestry system with production regulation by area relies on selective logging being conducted properly. In other words, this selection stage is crucial for ensuring that viable populations can remain in managed forests [35,76,77]. In addition, Romero [75] points out that the productive capacity of a managed area depends on the intensity of harvesting applied by the manager and the correct use of reduced impact harvesting techniques, resulting in direct implications for climate change mitigation.

It is widely recognized that biomass and floristic composition are naturally influenced by relief and soil type [24,28]. Based on forest inventories conducted by INPA in the 1980s, significant differences in basal area were observed because of relief variation. Samples taken on plateaus exhibited wood volumes of up to 210 m$^3$ ha$^{-1}$, with 155 different species and a density of 155 to 170 trees ha$^{-1}$, for DBHs greater than 25 cm. In contrast, samples in flat areas, but at lower elevations, showed values of 136 m$^3$ ha$^{-1}$, with 95 botanical species and a tree density of 135 trees ha$^{-1}$ [37]. The PROFLAMA, in an inventory also conducted in this area, observed a similar difference in volume [78].

Cluster analysis indicates that the topography of the site is a determining factor in the spatial arrangement and exploitation of the commercial species selected by the manager [79]. Hartemink [80] discusses soil geography and classification, including soil texture variation in different landscapes and its influence on the spatial distribution of species, ecological dynamics, and sustainable management of forest ecosystems. These results demonstrate a progressive differentiation in volume as one moves down the relief positions. Denser forests with a greater volume of larger trees are observed on the plateaus. Intuitively or not, logging followed this pattern throughout the area, aligning with the natural distribution of the stand.

To reinforce this, the study enabled a direct comparison between the conditions before and during logging. The graphs in Figure 16 illustrate that the forest before logging resembled the logged forest. This indicates that logging adhered to the natural distribution of the forest. (Figure 16).

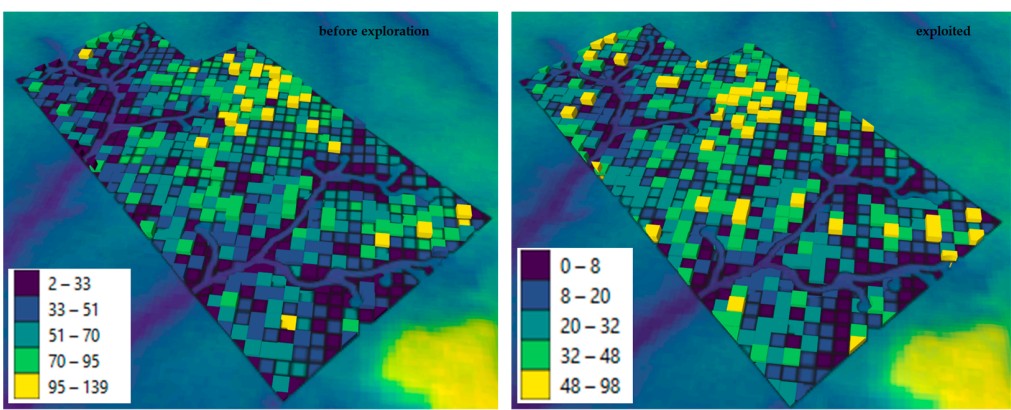

**Figure 16.** 3D representation of the volume before logging and of the trees actually logged (m$^3$ ha$^{-1}$).

Understanding that the Brazilian government has implemented forest management across more than 220 million hectares in the Amazon, it is crucial to emphasize the necessity of developing metrics and monitoring methods that consider the forest's heterogeneity and the specific distribution of individuals and volume in each area. This includes assessing the remaining stock to inform potential future interventions, such as cutting cycles [81–83].

It is important to note that relying solely on reduced impact logging (RIL) does not ensure the sustainability of forest management [84]. For timber production to truly contribute to conservation efforts, additional measures must be implemented [43]. Our methodology was instrumental in evaluating changes in the spatial distribution of commercial tree volume and abundance before, during, and after logging in concession areas. By utilizing simple inventory data and remote sensing imagery within a geographic information system, our approach shows promise in supporting effective and sustainable forest management in the Amazon.

Spatial and temporal analyses conducted within standardized cells, along with cluster analyses, proved to be effective in examining the spatial distribution and changes in forest structure within forest concession areas in the Brazilian Amazon. It is important to note that our study focused exclusively on public concession areas, as logging areas within private sustainable forest management plans were not included. This limitation should be considered, since public concession areas represent only a fraction of forest logging activities in the Amazon [85].

The study was centered on a single public concession area and may not fully represent logging practices across the entire Amazon region. Specifically, it focused on a particular area within FMU II (TUs 3 and 4) and exclusively examined spatial and temporal characteristics related to commercial volume and tree occurrence per hectare in aggregated form.

It is important to acknowledge that the data analyzed originate from a commercial inventory, focusing solely on species of economic value to the concessionaire. Therefore, it does not encompass the entire population of species within the area. Additionally, while the study area is situated in a region with a rich history of human presence, including communities such as quilombolas and indigenous peoples with deep connections to the forest, social and cultural factors were not incorporated into the analysis.

The expectation is that the findings of this study will encourage forest concessions to adopt the methodology proposed here, thereby aiding in the management of forest exploitation in the Amazon. Additionally, it is recommended that the methods employed to assess the impacts of logging and the forest's capacity for recovery, such as permanent plots, consider the insights provided by this study.

## 5. Conclusions

Logging within the forest management area in the study exhibits heterogeneity in terms of the spatial distribution of logging intensity, encompassing both commercial volume and abundance. In this study's conditions, logging intensity aligned with the natural

distribution of the forest, where the largest stocks were proportionally more exploited, and the greatest logging intensity occurred in areas with a greater stock of available timber. We recommend that managers consider this forest heterogeneity for forest exploitation using RIL techniques.

These findings hold significant implications for forest management and suggest constant monitoring in highly exploited areas and future improvements to current regulations to develop adaptive and more nuanced management strategies and foster the sustainability and conservation of forest resources.

**Author Contributions:** Conceptualization, A.H.M.O. and L.J.M.d.F.; methodology, A.H.M.O. and L.J.M.d.F.; formal analysis, A.H.M.O., L.G.M. and C.T.d.S.D.; investigation, A.H.M.O. and L.G.M.; resources, J.H.C.; data curation, A.H.M.O.; writing—original draft preparation, A.H.M.O.; writing—review and editing, A.H.M.O., L.G.M., J.H.C., M.M.M., C.T.d.S.D. and L.J.M.d.F.; visualization, L.G.M. and A.H.M.O.; supervision, A.H.M.O. and L.G.M.; project administration, A.H.M.O., L.G.M. and L.J.M.d.F. All authors have read and agreed to the published version of the manuscript.

**Funding:** This research received no external funding.

**Data Availability Statement:** Data are contained within the article and are also available from the corresponding author.

**Acknowledgments:** We thank the Brazilian Coordination for the Improvement of Higher Education Personnel (CAPES) for the scholarship granted to the first author. We would like to thank the Brazilian Forest Service (SFB) for providing the forest inventory data and PlanetLabs for providing the satellite images. In addition, we would like to thank the anonymous reviewers for their constructive suggestions, which helped to improve our paper.

**Conflicts of Interest:** The authors declare no conflicts of interest.

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
