# Peer review of "Spatio-Temporal Assessment of Heterogeneity by Logging Intensity in a Federal Concession Area in the Brazilian Amazon"

_forests, doi:10.3390/f15061062_

Round 1

Reviewer 1 Report

Comments and Suggestions for Authors

Find my comments in the pdf file and good luck.

Author Response

Dear reviewer,

Thank you very much for taking the time to review this manuscript. We appreciate the time and effort that you and the reviewers have dedicated to providing your valuable feedback on the manuscript. We are grateful to the reviewers for their insightful comments on the paper. We have been incorporate changes to reflect all of the suggestions provided by the reviewers. All the changes are highlighted in green and blue, and the correction control has been used to make it easier to see each adjustment made to the work. We have highlighted the changes within the manuscript. Please find the detailed responses below and the corresponding revisions/corrections highlighted/in track changes in the re-submitted files.

Point-by-point response to Comments and Suggestions for Authors

Comments 1: [Line 19: system]

Response 1: Thank you for comment and attention. The grammatical correction from "sistem" to "system" has been made in line 19.

Comments 2: I recommend to enlarge your introduction a bit more. For example, you need to mention what has been made so far in terms of monitoring/mapping of forest harvested areas, etc., using modern remote sensing approaches (satellites, UAVs, etc.) and GIS. Your work was based on Forest inventory and topographic data acquired from remote sensing so you need to add some information. Once you are referring to logging and GIS in combination with satellite images, then I strongly recommend to mention the following study by Abdollahnejad et al. 2019. You can also use it in the section of discussion… Here is the reference:

Abdollahnejad, A.; Panagiotidis, D.; Bílek, L. An Integrated GIS and Remote Sensing Approach for Monitoring Harvested Areas from Very High-Resolution, Low-Cost Satellite Images. Remote Sens. 2019, 11, 2539. https://doi.org/10.3390/rs11212539.]

Response 2: : Thank you for comment. The suggested article as well as the search for other references related to remote sensing and its applications for selective logging substantially improve the introduction of the work. We have included the suggested discussions  (See L49-77).

Comments 3: [Figure 1 is really interesting, very good job.]

Response 3: Thak you very much for this observation.

Comments 4: [Subsection 2.3 do you have some more detailed “picture” of the forest structure for the study area? Density based on basal area, or any diameter distribution graph, etc.?

Response 4 : Thank you for your suggestion. We have included a figure showing the spatialization of the trees and a graph with the diametric distribution (See L164).

Comments 5: [Lines 133-135: Indeed, and there are more works related to that. Strengthen your statement by adding this study too. Basically, the results showed that elevation is the most important topographic factor, for species and diversity distribution. Moreover, this paper revealed that the combination of topographic layers (particularly elevation) with QuickBird satellite data had the best results. Here is the reference: Abdollahnejad, A.; Panagiotidis, D.; Shataee Joybari, S.; Surový, P. Prediction of Dominant Forest Tree Species Using QuickBird and Environmental Data. Forests 2017, 8, 42. https://doi.org/10.3390/f8020042

Response 5 : Thank you for your comment. We have included the suggested article (see L171).

Comments 6: [Subsection 2.5. Have you considered using any other satellite sources, that could possibly result even better outputs, let`s say of higher spectral resolution like sentinel or pleiades (also considering the size of your study area) with higher spatial resolution (especially for selective logging areas) and to compare the results between them including planetscope? I do not disagree with the utilization of PlanetScope imagery; it aligns with the objectives and methodologies employed in your study. I am just wondering what was the main reason for that choice (besides the fact that they are free access data)?

Response 6: Thank you for your comment. That’s a good question. To explain further, as the results of the work are more focused on forest inventory data, Planet images were used secondarily more in the sense of evaluating the intensity of exploitation and the relationship between canopy openings and the volume exploited. And as these images have better spatial and temporal resolution than any other freely available sensor, and these are the most important characteristics for monitoring selective logging, it was decided to use them in isolation.

Comments 7: The findings and subsequent discussion are articulated and visually represented in a concise and comprehensive manner.

Response 7: Thank you for the positive comment.

Comments 8: Conclusions are clear and pretty straight forward as they should be.

Response 8: Thank you for the positive comment. 

Reviewer 2 Report

Comments and Suggestions for Authors

The objective of this study is listed as L71-74:  “To assess the importance of these factors, this study aimed to evaluate the spatial distribution of logging, changes in volumetric stock, and variations in the abundance of trees of commercial species in a SFM area following logging activities subjected to the Reduced Impact Logging (RIL) technique.”

However, it appears there is an unstated hypothesis that logging intensity (apparently in terms of percent?) would decrease in areas of highest wood stocks (see L24-25).  The purpose of the study needs to be updated to include any hypotheses testing.  L23-24: “Logging in the study area reveals…, contrary to expectations,…intensifying in areas with the highest wood stocks.”   I am a forester and I would think where there is more wood the logging intensity would increase, especially given the low amount of volume elsewhere not to mention the additional care needed around riparian area.

Romero et al.  (2021. Forest Management with Reduced-Impact Logging in Amazonia: Estimated Aboveground Volume and Carbon in Commercial Tree Species in Managed Forest in Brazil’s State of Acre. Forests 2021, 12, 481., not cited here and perhaps it should be) indicates that “Brazil requires management projects to conduct a “100% inventory” of the merchantable volume in the forest management unit, with the 100% inventory defined as the measuring and mapping of all individuals of commercial species, considering a minimum diameter at breast height (DBH) of 50 cm..” and “.. low-impact harvest techniques must be incorporated in the plans in order to minimize impacts.”   The study seems to indicate that the trees to be harvested are labeled in the initial inventory, but it is not clear if after harvest there is any actual checking between planned harvest and actual harvest.

Remero et al mention collateral logging damage, and discuss species of the tree as well.  The impacts of these two important topics should be more thoroughly discussed in terms of how they may contribute to the results.   In this manuscript L399-400 mentions  “The spatial distribution of commercial species and their preferences for different types of soil and terrain should be considered in further analyses”, however, given that a 100% ground inventory of certain sized diameter trees appears to be required, and which surely includes species name for the trees, it sounds as if that information already exists, so why not use it in this study?  

In terms of openings, perhaps there was collateral logging damage in the areas of greater volume, or perhaps the local managers wanted greater spacing in certain areas, or perhaps there was intentionally removal of too much volume in a specific hectare, or a combination or some other reason.  Can loggers remove logging-damaged trees?  Is there certainty if there is an opening that the stem of the tree no longer standing was actually removed?   This type of uncertainty should be considered for discussion in this study.  Lines 462-464 notes “…that the data analyzed originates from a commercial inventory, focusing solely on species of economic value to the concessionaire. Therefore, it does not encompass the entire population of species within the area.”  This sounds as if forest inventory field information used had only commercial species while the remote sensing information took into account all species? Please say more about this possible mismatch and effects on results.

Also, perhaps if the 1 hectare boundaries were located starting from a different point, the results would change greatly.  Could non-commercial trees have been removed through logging damage or some other reason and those removals are showing up on these maps?  Investigating these uncertainties is important in evaluating this technique.  L399 indicates that the average logging intensity was 28 m3/hectare which is below the maximum 30 m3/ha limit.  Perhaps in practice the average logging intensity is used to compare to the maximum 30 m3/ha rather than any specific hectare?

Can the remote sensing information provide an indication of the species of trees being removed?  If not, that may limit how much additional information this technique is providing.  It would be important to know if trees of rare species were being removed but this specific information does not seem to be available.

Please modify the text to address the questions above.  

Specifics:

L178-180 indicatesThe data was subsequently verified through visual interpretation. Thus, except for the individual pixels, all other data points were validated and aggregated…”   So was 100 percent of the data verified through field visual interpretation?  Please say more about this verification, and how accurate the spatialized data and maps are.

L14-15 in abstract needs to be rewritten because only people can consider; species do not consider.

Author Response

Dear reviewer,

Thank you very much for taking the time to review this manuscript. We appreciate the time and effort that you and the reviewers have dedicated to providing your valuable feedback on the manuscript. We are grateful to the reviewers for their insightful comments on the paper. We have been incorporate changes to reflect all of the suggestions provided by the reviewers. All the changes are highlighted in green and blue, and the correction control has been used to make it easier to see each adjustment made to the work. We have highlighted the changes within the manuscript. Please find the detailed responses below and the corresponding revisions/corrections highlighted/in track changes in the re-submitted files.

Point-by-point response to Comments and Suggestions for Authors

Comment 1: The objective of this study is listed as L71-74:  “To assess the importance of these factors, this study aimed to evaluate the spatial distribution of logging, changes in volumetric stock, and variations in the abundance of trees of commercial species in a SFM area following logging activities subjected to the Reduced Impact Logging (RIL) technique.”

However, it appears there is an unstated hypothesis that logging intensity (apparently in terms of percent?) would decrease in areas of highest wood stocks (see L24-25).  The purpose of the study needs to be updated to include any hypotheses testing.  L23-24: “Logging in the study area reveals…, contrary to expectations,…intensifying in areas with the highest wood stocks.”   I am a forester and I would think where there is more wood the logging intensity would increase, especially given the low amount of volume elsewhere not to mention the additional care needed around riparian area.

Response 1: Thank you very much for your comment. We've removed the phrase “contrary to expectations” (see L25) because it was really giving the impression that there was another hypothesis, when in fact we were referring to the current legislation in Brazil which limits logging intensity to 30 m³ ha-¹, i.e. it's not about the authors' expectations or the article's hypothesis, but rather the current legislation which in theory doesn't expect intensities greater than 30 m³ ha-¹. We agree that where there is a greater supply of wood, there will also be greater logging intensity, and this is exactly what our work points out.

Comment 2: Romero et al.  (2021. Forest Management with Reduced-Impact Logging in Amazonia: Estimated Aboveground Volume and Carbon in Commercial Tree Species in Managed Forest in Brazil’s State of Acre. Forests 2021, 12, 481., not cited here and perhaps it should be) indicates that “Brazil requires management projects to conduct a “100% inventory” of the merchantable volume in the forest management unit, with the 100% inventory defined as the measuring and mapping of all individuals of commercial species, considering a minimum diameter at breast height (DBH) of 50 cm..” and “.. low-impact harvest techniques must be incorporated in the plans in order to minimize impacts.”   The study seems to indicate that the trees to be harvested are labeled in the initial inventory, but it is not clear if after harvest there is any actual checking between planned harvest and actual harvest.

Response 2: Thank you very much for your comment. The article by Romero et al has been included in the discussions (See L429 and L447-450).

The check between the volume of timber planned for exploitation and the volume actually exploited is done indirectly by cross-checking tabular information, since the manager is obliged to declare the volume (forest credit) to be marketed, which must be close enough to the volume approved for exploitation in the management plan. In some cases, on-site inspections check the stumps of trees declared as logged.

Comment 3: Romero et al mention collateral logging damage, and discuss species of the tree as well.  The impacts of these two important topics should be more thoroughly discussed in terms of how they may contribute to the results.   In this manuscript L399-400 mentions  “The spatial distribution of commercial species and their preferences for different types of soil and terrain should be considered in further analyses”, however, given that a 100% ground inventory of certain sized diameter trees appears to be required, and which surely includes species name for the trees, it sounds as if that information already exists, so why not use it in this study?  

Response 3: Thank you for your comment. The study sought to relate abundance and volume to the intensity of exploitation, a simple and direct relationship. Future studies could delve deeper into the relationship between the intensity of exploitation and the ecological distribution of species (soil preference, habits, etc.), but this was not the aim of this study. However, in order to mention the possibility of collateral damage, we cite the study by Romero 2021 which also discussed this issue (See L429).

Comment 4: In terms of openings, perhaps there was collateral logging damage in the areas of greater volume, or perhaps the local managers wanted greater spacing in certain areas, or perhaps there was intentionally removal of too much volume in a specific hectare, or a combination or some other reason.  Can loggers remove logging-damaged trees?  Is there certainty if there is an opening that the stem of the tree no longer standing was actually removed?   This type of uncertainty should be considered for discussion in this study.  Lines 462-464 notes “…that the data analyzed originates from a commercial inventory, focusing solely on species of economic value to the concessionaire. Therefore, it does not encompass the entire population of species within the area.”  This sounds as if forest inventory field information used had only commercial species while the remote sensing information took into account all species? Please say more about this possible mismatch and effects on results.

Response 4: Thank you for your comment. Managers can only remove trees from the forest that are included in the management plan for extraction. The trees are labeled and georeferenced, so every opening in the forest canopy is associated with the species that have been logged. The remote sensing information therefore only reflects the species that were actually extracted, because as there are coordinates for each species, the visual validation on the images served to confirm the opening of the clearing due to the extraction of the trees and consequently the validity of the polygons, as exemplified in Figure 4 (see L208).

Comment 5: Also, perhaps if the 1 hectare boundaries were located starting from a different point, the results would change greatly.  Could non-commercial trees have been removed through logging damage or some other reason and those removals are showing up on these maps?  Investigating these uncertainties is important in evaluating this technique.  L399 indicates that the average logging intensity was 28 m3/hectare which is below the maximum 30 m3/ha limit.  Perhaps in practice the average logging intensity is used to compare to the maximum 30 m3/ha rather than any specific hectare?

Response 5: Thank you for your comment. We agree that the results could vary depending on different grid sizes, however, as it is a widely used unit and easy to compare with studies in other areas, we have assumed 1 hectare as the basic unit of analysis.

As mentioned above, the manager cannot remove any other trees from the forest that are not approved for extraction in the management plan. In other words, the remote sensing information is exclusively about the trees that have been harvested.

Comment 6: Can the remote sensing information provide an indication of the species of trees being removed?  If not, that may limit how much additional information this technique is providing.  It would be important to know if trees of rare species were being removed but this specific information does not seem to be available.

Response 6: Thank you for your comment. No, remote sensing is not able to indicate which specific species are being removed, in this study we focused on quantifying the canopy openings caused specifically by the toppling of trees in general. Indeed, information on the removal of rare species would be very important, but we don't have this information and unfortunately it's not possible to obtain it for remote sensing.

Specifics:

Comment 7: L178-180 indicates “The data was subsequently verified through visual interpretation. Thus, except for the individual pixels, all other data points were validated and aggregated…”   So was 100 percent of the data verified through field visual interpretation?  Please say more about this verification, and how accurate the spatialized data and maps are.

Response 7: Thank you for your comment. Yes, 100% of the data was verified through visual interpretation, because as we had the exact dates and coordinates of the trees that had been extracted, with the help of the image we were able to confirm the extractions and then quantify the size of the openings in the canopy. In this way, we were sure that each polygon classified as a clearing was associated with an extracted tree. (See L183 and L216-219)

Comment 8: L14-15 in abstract needs to be rewritten because only people can consider; species do not consider.

Response 8: Thank you for your comment: the text has been rewritten as follows: The volumetric distribution of native species and the intensity of logging often do not take into account the heterogeneity of the forest. (see  L15).

Round 2

Reviewer 2 Report

Comments and Suggestions for Authors

The objective of this study is listed as L100-103:  “To assess the importance of these factors, this study aimed to evaluate the spatial distribution of logging, changes in volumetric stock, and variations in the abundance of trees of commercial species in a SFM area following logging activities subjected to the Reduced Impact Logging (RIL) technique.”   

 Please reword or add text to this objective for clarity.  It is not clear what “these factors” are, and evaluation criteria are not clearly stated.  L394-396 states “that the main finding of this study is to demonstrate that the logging intensity… is not uniform and follows…”  So the purpose of the study was to demonstrate this result?    With the purpose written as it is, I would expect the conclusions to state that it is important to consider geospatial heterogeneity in forests when using RIL techniques, or when writing regulations regarding forest treatments, or something similar.  L25-26 in the abstract indicates results suggest the need to review management practices but it is not clear what management practices those would be or who needs to be the one reviewing those practices.  Please reword which man include adding text to the objective of the study so that the rest of the manuscript is consistent with the purpose.

 L98-99 indicates that the regulation does not take the heterogeneity into account so that managers can “exploit the forest without respecting its original structure and spatial distribution” but a regulation that doesn’t take into account the heterogeneity of the forest sounds like the regulation (and those who passed the regulation) are the ones not respecting the spatial distribution.  Please modify the text so there is consistent logic to the statements that are made. 

 Section 2.3 reads as though trees less than 40 cm in diameter were not included in the inventory data, and since forest inventory data were used to produce the volumes in the maps of cells in Figure 6 and 7, it sounds as though there may be more volume in this forest than is being reported, but it is in smaller trees.  Please modify the text to speculate how omitting these smaller trees in the analyses may be affecting the uncertainty of the results.

 L516-518 regarding the statement in the conclusion that “contrary to the forester’s anticipation that areas with the highest stock would be proportionally less logged…”   In order to say this in the conclusion please clearly document with references in the manuscript to a survey of foresters that shows the majority anticipates that areas with highest stock would be proportionally less logged.  Also, modify the text in the purpose of the study to be clear you are testing this hypothesis based on existing literature that foresters anticipate this.  Otherwise rewrite the text in the conclusions because there is nothing in the existing text that supports the inclusion of this statement. 

 Specifics:

L14-15 in abstract needs to be rewritten, the sentence as written does not make sense.  such as "Regulations involving the volumetric distribution... may not take into account the heterogeneity of the forest."  However, the specific sentence depends on the purpose of this study, which is not clearly defined.  The text describing the purpose of the study needs to be modified to be clear so that the rest of the text makes sense in relation to the purpose. 

Author Response

Dear Reviewer,

Thank you very much for taking the time to review this manuscript. We appreciate the time and effort that you have dedicated to providing your valuable feedback on the manuscript. We are grateful to the reviewers for their insightful comments on the paper. We have been incorporate changes to reflect all of the suggestions provided. We have highlighted the changes within the manuscript. Please find the detailed responses below and the corresponding revisions/corrections highlighted/in track changes in the re-submitted files.

Point-by-point response to Comments and Suggestions for Authors

Comment 1. The objective of this study is listed as L100-103:  “To assess the importance of these factors, this study aimed to evaluate the spatial distribution of logging, changes in volumetric stock, and variations in the abundance of trees of commercial species in a SFM area following logging activities subjected to the Reduced Impact Logging (RIL) technique.”  

 Please reword or add text to this objective for clarity.  It is not clear what “these factors” are, and evaluation criteria are not clearly stated.  L394-396 states “that the main finding of this study is to demonstrate that the logging intensity… is not uniform and follows…”  So the purpose of the study was to demonstrate this result?    With the purpose written as it is, I would expect the conclusions to state that it is important to consider geospatial heterogeneity in forests when using RIL techniques, or when writing regulations regarding forest treatments, or something similar.  L25-26 in the abstract indicates results suggest the need to review management practices but it is not clear what management practices those would be or who needs to be the one reviewing those practices.  Please reword which man include adding text to the objective of the study so that the rest of the manuscript is consistent with the purpose.

Response 1: Thank you very much for your considerations. Due to the relevant contributions to the manuscript, we have included the word "heterogeneity" in the title to better align with the objective (See L2). As suggested, we have rewritten the objective of the work to make it clearer and more concise (See L107-110).

To maintain the consistency of the conclusions, we have rewritten the paragraphs to incorporate your considerations (see L534-548). When we mentioned "contrary to the forester’s anticipation ...,” we were referring to the policymakers who expect the intensity of exploitation to be a maximum of 30 m³ per hectare, which in reality, it is not. To improve clarity, we have rewritten the sentence to suggest the need to consider forest heterogeneity for exploitation (See 537-540).

Regarding the suggestion to revise and reformulate the laws “revisiting management guidelines...” (see L545), we agree that our work is not focused on discussing legislation but rather on pointing out methods and techniques to support indicators aiming for sustainable management. Therefore, we have rewritten the sentence to “These findings hold significant implications for forest management, suggesting constant monitoring in highly exploited areas for future improvements to current regulations...” (see L543-545).

We have included a paragraph in the discussion to delve deeper into the criteria currently adopted for management and the need for a more refined analysis considering the heterogeneity of the area as well as which regulatory body in Brazil oversees this (See L433-438).

 We have rephrased the sentence in the abstract (L26-28) to clarify the importance of additionally considering forest heterogeneity alongside current rules to maintain sustainable management. We understand that it is important to monitor highly exploited areas to determine whether it is indeed necessary to change the current rules (See L409-412).

Comment 2. L98-99 indicates that the regulation does not take the heterogeneity into account so that managers can “exploit the forest without respecting its original structure and spatial distribution” but a regulation that doesn’t take into account the heterogeneity of the forest sounds like the regulation (and those who passed the regulation) are the ones not respecting the spatial distribution.  Please modify the text so there is consistent logic to the statements that are made.

Response 2: Thank you very much for the suggestion. The text in L100-102 has been modified to maintain a logical consistency throughout the discussions. The idea of the paragraph is to highlight the impact of considering the intensity of exploitation using a fixed and standard value for the entire area, disregarding heterogeneity on a more refined scale.

Comment 3. Section 2.3 reads as though trees less than 40 cm in diameter were not included in the inventory data, and since forest inventory data were used to produce the volumes in the maps of cells in Figure 6 and 7, it sounds as though there may be more volume in this forest than is being reported, but it is in smaller trees.  Please modify the text to speculate how omitting these smaller trees in the analyses may be affecting the uncertainty of the results.

Response 3: Thank you for your comment. The analyses were focused on evaluating only commercial trees that had potential for cutting, i.e., with DBH greater than 50 cm as required by the legislation. Since trees smaller than 50 cm could not be harvested, as well as trees within PPAs, these trees were excluded from the analysis. We have included this information more clearly in L171-173.

In this way, the study focused on evaluating only commercial trees available for cutting outside the limits of PPAs and with a DBH greater than 50 cm. It is important to note that the forest inventory for commercial purposes does not consider all species in the area and focuses only on species that have some commercial value for the manager (See L518-520).

Comment 4. L516-518 regarding the statement in the conclusion that “contrary to the forester’s anticipation that areas with the highest stock would be proportionally less logged…”   In order to say this in the conclusion please clearly document with references in the manuscript to a survey of foresters that shows the majority anticipates that areas with highest stock would be proportionally less logged.  Also, modify the text in the purpose of the study to be clear you are testing this hypothesis based on existing literature that foresters anticipate this.  Otherwise rewrite the text in the conclusions because there is nothing in the existing text that supports the inclusion of this statement.

Response 4: Thank you very much for the suggestion. As mentioned in response 1, to maintain consistency, we have excluded the mentioned paragraph and rewritten it to incorporate your considerations (see L543-545). When we mentioned "contrary to the forester’s anticipation ...," we were referring to the policymakers who expect the intensity of exploitation to be a maximum of 30 m³ per hectare, which in reality, it is not. To improve understanding, we have rewritten the sentence to suggest the need to consider forest heterogeneity for exploitation and suggesting constant monitoring in highly exploited areas to assess future improvements to current regulations (See 543-545).

 Specifics:

Comment 5. L14-15 in abstract needs to be rewritten, the sentence as written does not make sense.  such as "Regulations involving the volumetric distribution... may not take into account the heterogeneity of the forest."  However, the specific sentence depends on the purpose of this study, which is not clearly defined.  The text describing the purpose of the study needs to be modified to be clear so that the rest of the text makes sense in relation to the purpose.

Response 5: Thank you for the suggestion. We have rewritten the text to align the sentence with the proposal of the paper (see L16-17).